# Students’ Wellbeing during Transition from Onsite to Online Education: Are There Risks Arising from Social Isolation?

**DOI:** 10.3390/ijerph18189665

**Published:** 2021-09-14

**Authors:** Gina Ionela Butnaru, Alina-Petronela Haller, Larisa-Loredana Dragolea, Alexandru Anichiti, Georgia-Daniela Tacu Hârșan

**Affiliations:** 1Department of Management, Marketing and Business Administration, Faculty of Economics and Business Administration, Alexandru Ioan Cuza University of Iași, 700505 Iași, Romania; gina.butnaru@uaic.ro; 2“Gheorghe Zane” Institute for Economic and Social Research, Iași Branch of Romanian Academy, 700488 Iași, Romania; geo_tacu@yahoo.com; 3Department of Business Administration and Marketing, Faculty of Economics, 1 Decembrie 1918 University of Alba Iulia, 510009 Alba Iulia, Romania; larisadragolea@uab.ro; 4Department of Business Administration, Ștefan cel Mare University of Suceava, 13 University Street, 720229 Suceava, Romania; alexandruanichiti@gmail.com

**Keywords:** satisfaction and wellbeing, pandemic stress, COVID-19 pandemic, online education, social isolation

## Abstract

The COVID-19 pandemic has caused disruption to activities in many fields, including education and lifestyle. Major changes have taken place in the education system, where specific activities migrated suddenly from onsite to online. As a result, this period has witnessed an increased interest in impact studies that analyse the perceptions of the actors involved in the educational process. Based on the survey data (*N* = 665), the perceptions of the students in Romanian universities with regard to the effects of online education during the pandemic on their wellbeing were analysed. The empirical apparatus—SEM analysis—reached the following conclusions: the students’ wellbeing was increased under the traditional education system; the economic crisis has caused concern, and a decrease in their wellbeing; their contamination fear is moderate to low, and does not influence their wellbeing; they have been discouraged in terms of their personal development during the pandemic, and their wellbeing has suffered as a result; the role of institutions is extremely important, given that the students’ ability to study online depends on the universities’ efficiency in implementing the online system.

## 1. Introduction

The COVID-19 (Coronavirus Disease 2019) pandemic has created social disturbance, and changed several societal norms, causing stress and forcing lifestyle changes. The measures taken by authorities to reduce contagion have helped greatly in reaching the needed goals; however, people’s wellbeing has been affected [1]. Social isolation, insecure jobs, and limited social interaction have accentuated the state of insecurity and anxiety of young people and young adults [2]. During this time, anxiety, which reduces wellbeing and affects people’s mental health, can become chronic and burdensome as an adaptive response to the threat of the pandemic [3]. Most of the concerns relate to personal health and that of loved ones, but also to foreseeable economic issues. According to Asmundson et al. [4], the highest risks are associated with pre-existing mental sensitivities, and the factors that reduce wellbeing while affecting mental health are fear of contagion, socioeconomic consequences, and traumatic stress.

The pandemic has increased anxiety and altered individual behaviour [5]. Young people’s wellbeing has been affected by fear of contagion, and even more by unexpected isolation and online learning. Young people who did not make keeping informed on the status of the pandemic a priority, as well as older people, tend to be in a better state [6], while the wellbeing of women and young people in non-medical studies is on a declining trend [7].

The pandemic has affected mental health—especially that of people who are lacking in job security [8] or subject to major risks [9], related either to losing one’s job, or to the impossibility of finding one, including for young people who are still in school. The COVID-19 pandemic has affected young people’s mental health in different ways, but social isolation and the almost complete loss of all social activities, school, work, and training also affect them in the short term [10]. Forecasts of a large-scale economic crisis can affect young people’s psychological wellbeing. Those who are vulnerable include emigrants, young people, women, and poorly educated people who suffer from labour market changes, affecting their wellbeing and their mental health [11]. The COVID-19 pandemic is a threat to the wellbeing of young people and their families because of social disturbances—especially financial insecurity, the need to take greater care of their health, and not knowing when the pandemic will end—and the consequences will be long term [12]. Issues related to psychological wellbeing have been a concern since the 2008 crisis, and have seen a resurgence with the COVID-19 pandemic because of concerns caused by changes in the labour market, in spite of the fact that not all members of society have been equally affected [13]. The psychological impact of the pandemic has led the authors to focus on young people’s perceptions of the current changes. Young people in training have faced new challenges. The changes to the educational system, along with the insecurity and risks implied by a totally new situation, have motivated the authors to analyse the effects of online education on students’ wellbeing, under the isolation conditions imposed by the COVID-19 pandemic.

The COVID-19 pandemic is, and will be for a long time, a major challenge for all institutional structures, including the education system. As with teleworking, online education has a strong impact on students’ mental health and wellbeing. Pandemic stress is increasingly present in students’ lives: inability to adapt quickly enough to online teaching and learning; lack of socialisation; insecurity; sedentary lifestyles; increasing the time dedicated to online activities; simultaneous work, learning, and daily living in the same place; more rigorous programme compliance; an avalanche of fake news; and almost completely giving up on age-specific social habits. All of the above will have a negative impact on young people’s wellbeing, especially in terms of emotional recovery, anxiety, and educational performance, and there is a need for constant involvement of counsellors and psychologists [14,15].

Our main purpose is to analyse, based on the survey data (*N* = 665), the perceptions of the students in Romanian universities of the effects of online education during the pandemic on their wellbeing. We designed seven research hypotheses. The central concept is that of wellbeing, by which we understand self-esteem, self-determination, resilience, quality of life, good mood, and good mental health [16]. The results outline a preliminary picture of the educational impact of the pandemic through online learning, and they are only the first step in a broader study that the authors will carry out in the future.

Education and the related systems are among the most affected by the migration from onsite to online. The prospects of returning to onsite learning are low because of the risk of contagion [17,18,19,20]. Previous studies reveal a high degree of insecurity regarding the near future of the education process [21,22,23,24]. The pandemic has had significant effects on education [25,26,27,28,29], with the institutions in the field being forced to adapt to it. Romania has taken measures to control the pandemic: Schools, high schools, and universities have gone online. Online education and training require flexible teaching and learning, and courses can be easily accessed. The paradox of this crisis is the amazing and unprecedented speed of transferring courses from onsite to online [30,31].

This study is a comparative analysis of the effects on the wellbeing of students from universities in Romania in the context of the online education imposed by the COVID-19 pandemic. The perceptions of students and pupils on the effectiveness of online courses have been studied intensely by Bentley [32], Bali and Liu [33], and Platt, Raile, and Yu [34]; however, they all used a hybrid scenario, with both onsite and online courses. The novelty of this research is that it analyses the impact of the pandemic on the wellbeing of the students studying for bachelor’s and master’s degrees, where only online courses were available.

This study focuses on the following issues: students’ wellbeing; their willingness to study in the traditional way; how they relate to online education; fear of contagion; the role of universities in creating a safe environment; students’ perceptions of the impact on their career and personal journeys; the evolution of the education system; and the economic implications of the COVID-19 pandemic.

## 2. Literature Review and Research Hypotheses

### 2.1. Contagion Fear and Wellbeing

The pandemic has generated a growing interest in impact studies that analyse the relationship between the perception of contagion fear and wellbeing. The threat of the pandemic has generated insecurity, fear, stress, vulnerability, and concern for the future, with negative impacts on wellbeing [35].

One of the main causes of anxiety in young people is fear of contagion, both for themselves and for their loved ones, as shown by Odriozola-González et al. [36] in a study conducted on Spanish students. The concern for the health of family and friends is a stress factor for Swiss students [37]. Tanga et al. [38] explored the prevalence of stress and depression in a group of Chinese students in quarantine from six universities in order to identify the mental risk factors, showing that the most important risk factor for mental suffering is extreme fear.

Taylor et al. [39] analysed the consequences of quarantine during the pandemic in a group of Canadians and a group of Americans; general stress, combined with isolation and an attempt to avoid contagion, were noted. The acute suffering during the social isolation was caused by the fear of contagion, including in relation to loved ones, and the lack of control over such a situation.

An analysis carried out by Tee et al. [40] has shown that 16.3% of respondents perceived the psychological impact of the COVID-19 pandemic as moderate to severe; for 16.9%, the depressive symptoms were moderate to severe; for 28.8%, the anxiety level was moderate to severe; for 13.4%, the stress level was moderate to severe. Cao et al. [41] consider that the COVID-19 pandemic has generated an unbearable psychological pressure; the authors found that approximately 25% of students under study had low-to-severe anxiety. Anxiety is also caused by economic instability and delays in academic activities. Bitan et al. [42] showed that gender, socio-demographic characteristics, chronic diseases, belonging to a risk group, and death caused by COVID-19 in the family or in the social group were all positively correlated with fear of contagion, and produced anxiety, stress, and depression.

Martínez-Lorca et al. [43], studying a group of Spanish students with a mean age of 21.59, noted that they experienced fear and anxiety, but at a moderate rather than a high level.

Fear of contagion is one of the main predictors of anxiety and depression, according to Ahorsu et al. [44], Pedrosa et al. [45], Pakpour et al. [46], Schimmenti et al. [47], Olaimat et al. [48], Yehudai et al. [49], and many others.

### 2.2. The Economic Crisis Caused by the COVID-19 Pandemic, and Its Impact on Wellbeing

As stated above, the COVID-19 pandemic has affected and continues to affect the psychological state and the wellbeing of many people [35]. Ensuring social and economic efficiency depends precisely on psychological wellbeing. World governments have taken preventive measures, such as the quarantine of the population, wearing protective masks and gloves, lockdowns [50], online education, and working from home. However, social distancing, isolation, and travel restrictions have reduced the labour force in all fields, and have caused great losses of jobs, resulting in a strong fear regarding the imminence of a significant economic crisis [51]. Buheji et al. [52] note a contraction of the global economy by 12.5% in the first half of 2020. This could get worse, especially in developing countries, where economic recovery will be slow. Economic pressure is triggered by inflation and unemployment [52], which also results from people’s inability to participate in economic and social life. The UNDP (United Nations Development Programme) [53] argues that the unemployment issue is due to the decrease in economic activity, as a result of the suspended production processes, and because of the effects of recession on global welfare. Brodeur et al. [54] showed that the COVID-19 pandemic has caused a decrease in the global economy, as well as significant insecurity among the population, mostly due to unemployment.

Flanagan et al. [55] argue that the disastrous impact of the COVID-19 pandemic is aggravated by the unsustainability of economic globalisation, caused by the imbalances between supply and demand, and the inequality in the labour market, which mainly affects young people, women, and emigrants.

In a study involving respondents with a mean age of 36, Barrafrem et al. [56] noted their very pessimistic attitudes towards their economic situation and wellbeing, with the most optimistic attitudes being found among the highly educated population.

Nicola et al. [51] showed that 900 million children and students have been affected by the closure of educational facilities, with large-scale social and economic implications including the impossibility of providing free meals to children from low-income families, and school dropout due to the lack of technology for online courses. Post-university research was the most affected in tertiary education, because many topics contributing to economic growth and progress were put on hold. Above all, however, the negative effects on pupils’ and students’ wellbeing should be noted.

### 2.3. Is Wellbeing Influenced by Face-to-Face or Online Education?

Students’ wellbeing is linked to emotional resilience and a healthy lifestyle [57]. It is very important that educational institutions consider the wellbeing of pupils and students so that they will be able to make healthy lifestyle choices, and to understand the importance of such choices for their wellbeing [58]. Students with low levels of wellbeing are more likely to suffer from depression, anxiety, and stress [59]. Until the start of the COVID-19 pandemic, traditional education was predominant compared to online education all over the world. However, concerns about the efficiency and timeliness of online education have existed for a very long time, as shown by Barrett [60]. Many organisations have formed virtual work teams to collaborate on a variety of tasks, and although the performance and effectiveness achieved have been similar to the performance achieved by traditional teams, face-to-face team members have reported higher levels of satisfaction [61]. One of the most important advantages of online education is providing more flexibility to the time and space used for learning [62]. Nevertheless, the students’ perception of their own wellbeing includes an important emotional component.

The pre-pandemic literature contains relevant studies on students’ wellbeing and satisfaction in relation to online education [63]. Before the pandemic, online education was a personal option; now, for the majority of higher education institutions, it is the only possibility. Therefore, a higher level of skill in using electronic devices increases student satisfaction with online courses [64]. Furthermore, the interface of the online platform, customised for users, represents an element of satisfaction and improves students’ wellbeing [65]. The increase or decrease in the satisfaction level and, therefore, in the impact on students’ wellbeing, depends on multiple factors [66], and face-to-face learning is one of the most important of the aspects that contribute to students’ wellbeing. Although online education also has advantages, there is still the need for direct interaction among students, and between students and teachers, through facial expressions, gestures, impact, interaction, feedback, connection, and wellbeing. In order to increase wellbeing in the context of online education, the focus of universities will have to be on how to facilitate social–emotional learning in virtual classrooms. Thus, educational institutions should adapt rapidly, and should identify solutions so that the students will be able to achieve wellbeing, even if they will not be able to refer to a model [67]. A UNICEF (United Nations Children’s Fund) report [68] shows that online courses should take place according to a well-organised plan; otherwise, they represent an additional stress for pupils, with negative consequences linked to their mental health and wellbeing.

### 2.4. Online Education and Students’ Personal Development

Students’ personal development can best be realised during the learning process; it has two aspects: one refers to the responsibility to help students make the most of any opportunity to learn, in a formal or informal manner. The second refers to shaping the framework for self-directedness in learning, i.e., promoting a culture of responsibility for one’s own performance and learning [69].

Universities are constantly concerned with students’ personal development; they try to support students with complementary services, in addition to educational ones. Such are the information, career consultancy and orientation, and personal and professional development services. Most experts consider the informal interactions between students and academic staff to be directly linked to the increase in the former’s personal development and academic results [70]. University courses have been adapted, and now take place mostly online, with the purpose of supporting students’ personal development and ensuring its continuity; in turn, personal development has also adapted.

Today, career consultants play a significant role in students’ lives, as they help students to develop their potential [71]. Some university centres hold daily meetings on various platforms, where consultancy and personal development services are provided so as to increase wellbeing and engagement in activities that further students’ personal development [72]. The sudden switch to online learning may seriously affect the careers of university graduates [73]. Personal development is essential in students’ careers. Face-to-face learning and counselling are preferred by students for communication purposes, in which a shared understanding has to be derived, or in which interpersonal relations must be established [74].

### 2.5. The Link between Personal Development and Wellbeing in the Context of Online Learning during the COVID-19 Pandemic

Depending on the educational process, the UNDP [53] draws attention to the fact that although schools are not closed all over the world, and online learning is the new standard at present, there is, however, a steep decline in human development and learning efficiency. Thus, 9 out of 10 students are affected by the current crisis, and this cancels out the progress made in human development over the past six years. A report from the World Bank [75] highlights the impact of the COVID-19 pandemic on the financing of the education system in terms of reducing and, at best, maintaining its funding level, with negative impacts on learning, university enrolment rates, and educational services in general. The COVID-19 pandemic also threatens public education, posing fragmentation risks, because some of the teachers and students will not return to schools when they reopen [76]. Moreover, these unprecedented measures have caused great disturbance all over the world. The mental health and the psychological wellbeing of children and young people have been highly prejudiced, with long-term consequences for their wellbeing. The report from ECLAC–UNESCO (Economic Commission for Latin America and the Caribbean—United Nations Educational Scientific and Cultural Organisation) [77] shows that the current situation and the new educational methods seriously affect mental health and personal development, as well as exposing children and teenagers to violence.

A study conducted on a group of students in Malaysia has shown that the current situation generates a strong feeling of anxiety deriving from financial constraints, online learning, and insecurity about academic performance and career perspectives [78]; this analysis reflects the pandemic’s negative influence on the psychological wellbeing of pupils and students, leading to depression and anxiety, and the increased insecurity and negative influence on students’ academic progress could have an impact on their personal development and wellbeing in the long run. The authors note that students living alone have a higher degree of anxiety than those who live with their families or friends, as the company makes them feel safe and secure. Moreover, the online examination method is another stress factor, and is correlated with students’ personal development. The lack of practice in taking online exams, along with the stress caused by various factors (stable Internet connection, examination method, the examiner’s ability to understand and empathise, the ability to use online platforms, respecting the time frame, including sending/attaching the exam sheet, etc.), can generate negative, unpredictable and massive changes in daily routines and their mental health [79].

In addition to theoretical knowledge and practical skills, educational institutions should provide personal development information to the same extent, and focus on the students’ health and wellbeing [80]. Individual wellbeing is one of the most important goals pursued by individuals, and it derives from the basis of individual personal development [81]. Positive variations in wellbeing can be achieved via both the easier integration of young graduates into the labour market and, directly, in the proper management of feelings and reactions under stress, including the pandemic, panic, feelings of isolation, and other negative feelings.

Odriozola-González et al. [36] studied the mental wellbeing of Spanish students during the COVID-19 pandemic, and found a relatively high percentage of students who showed symptoms ranging from moderate depression to extremely severe anxiety, with higher incidence among students compared to the general population. Among the causes of these symptoms, the following were identified: fear of contagion, familial economic issues, disruptions in the educational environment, the effects of the pandemic on education and jobs, and reduced social interactions. Elmer et al. [37] analysed the activity of students on social networks, and compared the mental states of Swiss students before and after the pandemic. Thus, it was found that interaction on social networks decreased, students stopped studying together, and the levels of stress, anxiety, loneliness, and depressive symptoms increased. The most important stressors were lack of social life and concerns related to health, family, and friends, but also concerns about their future—that is, the economic impact, the impact of the new educational environment on their academic performance, and reduced opportunities in the labour market.

Aristovnik et al. [82] analysed a group of students from 62 countries, concluding that once COVID-19 measures were applied, they were mainly concerned with issues related to their future professional careers, but at the same time they experienced boredom, anxiety, and frustration. Some students were less satisfied with the academic activity during the pandemic, and others faced financial difficulties. Emotional issues were especially important for female students.

### 2.6. The Efficiency of Universities after Switching to Online Education

Higher education institutions have had to adapt to the new learning requirements. Thus, online applications, learning platforms, and educational resources have been created to ensure students’ communication and educational continuity [83,84]. Online courses were originally designed to be taught in classrooms, with information transmitted and stored using technology, so that they could later be accessed repeatedly by students. There is an important positive relationship between a student’s repeated accessing of a course and their level of knowledge and thinking; therefore, the course materials—uploaded or stored—will need to be developed and improved [85]. Because of the easy access to information, students appreciate the online learning style. Basilaia and Kvavadze [83] confirmed that the rapid transition from face-to-face education to online education had been successful, and that the experience gained could be used in the future. Moreover, the experiences during the pandemic will determine the ability of future generations to adapt to new laws, regulations, online study platforms, etc., and to find solutions to future challenges [22,86,87,88,89,90,91,92].

Studies show both positive and negative opinions of students regarding online learning [93,94,95]. There were positive opinions about the flexibility, profitability, and research perspectives in the online environment; Internet network accessibility; and the interfaces of the platforms used. There were negative opinions about teachers’ delayed feedback; the lack of immediate technical support; the lack of self-regulation and motivation; monotonous, boring teaching methods; the strong sense of isolation; and poorly designed and unattractive teaching content. The universities struggled and, as a result, time management skills have been improved, along with the technical knowledge required for the use of educational platforms, and the accessibility of online platforms provided for each student [96].

### 2.7. The Present Study

Students’ wellbeing represents one of the strategic goals of universities. Depending on the university’s material, financial, or human resources, the service quality management also investigates the level of student satisfaction reflected in students’ wellbeing. The unexpected transition to online education, without prior preparation for either teaching or student evaluation, has affected the students. The effects of the transition from onsite to online learning can only be captured over time, and will require much research. It is necessary to identify both students’ fears and the associated risks in order to implement appropriate measures and to counteract the negative effects of future periods of isolation.

Our goal was to analyse the perceptions of Romanian students regarding the effects of online education during the COVID-19 pandemic, from the perspective of their wellbeing. We started from seven hypotheses, and created a theoretical model based on data collected by survey. The model is shown schematically in Figure 1.

According to this model, the hypotheses are:**Hypothesis 1****(H1).** Fear of SARS-CoV-2 contagion has a negative effect on students’ wellbeing;**Hypothesis 2 (****H2).** The perception of a future economic crisis generated by the COVID-19 pandemic has a negative effect on students’ wellbeing;**Hypothesis****3 (H3).** The desire to study face-to-face has a negative impact on students’ wellbeing;**Hypothesis 4****(H4).** The migration from face-to-face learning to online learning causes students to suffer from anxiety, because it is associated with negative perceptions of their personal development;**Hypothesis 5****(H5).** A negative perception of personal development has a negative effect on students’ wellbeing;**Hypothesis 6****(H6).** The ease of studying online is positively correlated with the perception of universities’ efficiency;**Hypothesis 7****(H7).** The positive perception of universities’ efficiency has a positive effect on students’ wellbeing.

Both validated and invalidated hypotheses have led to conclusions that are useful for further research on young people’s wellbeing, and whether and how online education is responsible for isolation. The hypotheses refer to students’ perceptions of the following: fear of contagion; the economic crisis caused by the COVID-19 pandemic; the efficiency of face-to-face education; the feelings of isolation and anxiety caused by the pandemic; universities’ ability to adapt the teaching process to the requirements imposed by the pandemic; and universities’ efficiency in ensuring moral support so that students, uncertain of their personal or professional future, manage to avoid the feeling of isolation.

The conclusions draw attention to young people’s perceptions of a new but perhaps repeatable situation. The COVID-19 pandemic is not the only reason for online education. In the digital era, the education system could also be digitised. In the future, a hybrid learning system could be implemented. Our study can help policymakers and educational institutions to understand students’ perceptions of physical isolation and, thus, adopt educational strategies that not only include online teaching, but also streamline it to achieve optimal outcomes for all stakeholders.

The results of this study will be complemented by future research that will identify ways in which students can adapt to unforeseen situations in order to avoid the feeling of isolation, and to improve their academic performance.

## 3. Methodology and Methods of Research

### 3.1. Description of the Research Method

A questionnaire-based survey was used as a data collection tool. The questions contained items measured with a 5-point Likert scale (1–5). The study subjects were bachelor’s degree and master’s degree students from Romanian universities. Respondents were recruited by sending e-mails or online messages in study groups. E-mails were sent to all students from the following specialisations: Business Administration (undergraduate), Economy of Commerce, Tourism, and Services (undergraduate), International Relations and European Studies (undergraduate), Marketing (undergraduate), Tourism Geography (undergraduate), Business Administration in Commerce, Tourism, and Services (graduate), Tourism Management (graduate), Tourism and Hotel Management (graduate). In total, 1182 e-mails were sent. The beneficiaries interested in answering the questions accessed the link to the online survey in Google Forms. In total, 665 respondents completed the questionnaire, which represents approximately 56% of the total population of the specialisations.

### 3.2. Sample

There are various recommendations with regard to the size of a representative sample when performing structural equation modelling. This research followed the suggestions of Comrey and Lee in 1992, quoted by Field [97] and Butnaru et al. [98], on the suitable size of a representative sample: 100 = poor, 200 = satisfactory, 300 = good, 500 = very good, 1000 and beyond = excellent. The analysed population included 665 respondents, all of whom were students of universities in Romania. All of the subjects had the opportunity to ask questions or express their concerns regarding the survey.

Table 1 presents the sample’s descriptive statistics. Of the 665 subjects, 535 were female students (80.45%), and 130 were male students (19.55%). In terms of education, 508 were bachelor’s degree students and 157 were master’s degree students.

### 3.3. Method

Structural equation modeling (SEM)—a set of statistical techniques—is a comprehensive method of validating a model of latent constructs. SEM includes:(a)Exploratory factor analysis;(b)Confirmatory factor analysis (measurement model);(c)Estimation of the relations among the latent factors (structural model);(d)Validation of the model.

The steps of the structural equation modelling are displayed in Figure 2, and follow the recommendations of Dragan and Topolšek [99]. The statistical software used for all stages was Stata version 13 (StataCorp, College Station, TX, USA).

## 4. Statistical Analysis

### 4.1. Exploratory Factor Analysis

Dragan and Topolšek [99] used an exploratory factor analysis to identify the latent factors. We employed this technique to identify the underlying relationships between measured variables. After the main analysis, eight factors with an eigenvalue above 1 were obtained, meaning that there are probably eight basic factors (latent variables) in the questionnaire (Table 2).

An eigenvalue represents the proportion of variance explained by the component. Kaiser’s criterion [100] specifies that, for the analysis, only components with a value of 1.0 or bigger should be retained. Kaiser’s criterion is the default retention method in many statistical packages, including Stata. Therefore, given that there are eight factors with an eigenvalue > 1, the orthogonal factor rotation procedure (Kaiser’s varimax) was followed. Stevens [101] recommends that the questions with factors with loading over 0.400 be retained. Cross-loading is when a variable is loaded on more than one factor.

### 4.2. Confimatory Factor Analysis

After analysing the exploratory factors, the research model was proposed, and the condition was that the standardised loadings be at least 0.400. The estimated measurement model was then indicated using confirmatory factor analysis (CFA).

The validity property of the indicator elements in the model was tested using the CFA technique. CFA has wide applications, especially in the field of scale development and construct validation. Moreover, the strength of this method lies in the ability to allow the correlation of error variations to minimise the difference between estimated and observed matrices [102,103]. The evaluation of how the model fits the data was carried out using multi-criteria indices including the chi-squared (χ^2^), the normed chi-squared (χ2/df), and the comparative fit index (CFI). The Tucker–Lewis index (TLI) and the root-mean-square error of approximation (RMSEA) were also measured.

Matching quality measures were used to assess the CFA model. For the proposed model, the RMSEA is 0.064, the CFI is 0.917, the TLI is 0.916, and the SRMR is 0.068.

The minimum acceptable values according to Dragan and Topolšek [99] are shown in Table 3.

Composite reliability was applied to test the degree to which the indicator variables converge and share the proportion of variance. The CR (construct reliability) value varies between 0 and 1, and a higher value implies a higher level of item reliability [103]. A cutoff point of 0.7 or higher for CR is required to establish that the indicator elements are reliable, and that they share a large variance with the latent construct.

### 4.3. The Estimation of the Relationships between the Latent Factors and the Model Validation

Matching quality measures were used to assess the general structural model. The RMSEA is 0.064, the CFI is 0.906, the TLI is 0.905, and the SRMR is 0.078. Therefore, given the results, the model is a good fit.

SEM analysis was used to test the seven hypotheses proposed in this study.

The standardised regression coefficients show that H3, H4, H5, H6, and H7 are validated, while H1 and H2 could not be validated. The coefficients are shown in Table 4.

The standardised coefficients are shown in Figure 3.

In this study, we used structural analysis (SEM), which tested the following hypotheses:**Hypothesis 1****(H1).** Fear of SARS-CoV-2 contagion has a negative effect on students’ wellbeing. Path analysis showed a statistically insignificant effect of the fear of viral contagion on the students’ wellbeing. Therefore, H1 is not validated. Our results do not support the existence of a positive relationship between fear of contagion and young people’s wellbeing;**Hypothesis 2 (****H2).** The perception of a future economic crisis generated by the COVID-19 pandemic has a negative effect on students’ wellbeing. This hypothesis was tested using the structural model, and the results show that there is a statistically insignificant positive relationship between the two variables; therefore, H2 is not validated;**Hypothesis****3 (H3).** The desire to study face-to-face has a negative impact on students’ wellbeing. This hypothesis was tested using the structural model, and the results show that there is a statistically significant relationship between students’ desire to study onsite and their wellbeing (β = 0.337, p = 0.000). Thus, H3 is validated. The traditional education system favours students’ wellbeing;**Hypothesis 4****(H4).** The migration from face-to-face learning to online learning causes students to suffer from anxiety, because it is associated with negative perceptions of their personal development. This hypothesis was tested via SEM analysis, and was validated. Migrating from the traditional to the online learning system causes students to have negative perceptions regarding their personal development and wellbeing;**Hypothesis 5****(H5).** A negative perception of personal development has a negative effect on students’ wellbeing. The results of SEM analysis confirmed a statistically significant effect of the negative perception of personal development, which means that a stronger negative perception of personal development will have a negative effect on students’ wellbeing. The conclusion is that students are pessimistic regarding their personal development when the traditional education system changes, and this affects their psychological wellbeing;**Hypothesis 6 (H6).** The ease of studying online is positively correlated with the perception of universities’ efficiency. Results show a positive correlation of the ease of studying online with the perceived efficiency of the university (*β* = 0.554, *p* = 0.000). Hypothesis H6 is therefore validated;**Hypothesis 7 (H7).** The positive perception of universities’ efficiency has a positive effect on students’ wellbeing. Results show that a positive perception of the university’s efficiency will decrease the levels of stress and anxiety in students. Therefore, it has a positive effect on their wellbeing. Hypothesis H7 is thus validated.

The ability of universities to effectively apply online learning methods is reflected in students’ ability to learn easily, which gives them a positive perception of the institutional efficiency and a sense of wellbeing.

## 5. Discussions

Our goal was to analyse the perceptions of Romanian students of the effects of online education during the pandemic on their wellbeing. Our findings stem from the seven hypotheses formulated in accordance with the theoretical model that we chose. The hypotheses were designed in correlation with a series of supporting variables identified as factors, and established in accordance with the literature, the current situation, and the forecasted trends. Thus, two important objectives have been achieved: the results obtained are statistically significant and useful; the study enriches the literature in the field, and represents a solid starting point for future research. The research hypotheses derive logically and realistically from the analysis and synthesis of the following topics: contagion fear and wellbeing; the economic crisis caused by the COVID-19 pandemic, and its impact on wellbeing; the influence of onsite or online education on wellbeing; online education and students’ personal development; the relationship between personal development and wellbeing in the context of online education during the COVID-19 pandemic; and universities’ efficiency after switching to online education.

The hypotheses were a powerful tool that guided the entire scientific approach. Analysing the values obtained, and testing the validity of the model, we arrived at results comparable to those of other, similar studies or, conversely, different results.

Our results contradict the findings of Paredes et al. [35], Taylor et al. [39], and Cao et al. [41], but support those of Orizola-González et al. [36], Tee et al. [40], and Martínez-Lorca et al. [43]. Fear of contagion is still present, and universities must take it into account when designing their teaching strategies. In response to this fear, many universities around the world plan to simultaneously organise both onsite and online courses in order to meet students’ requirements and streamline the entire educational process.

An economic crisis is cause for concern, including among young people who are still studying, resulting in a decline in their wellbeing. Our results differ from those of Nicolaa et al. [51] and Barrafrem et al. [56], but are similar to those of Peterson and Tankom [104].

The hypothesis by which we tested the negative impact of students’ desire to study face-to-face was validated, and was consistent with the results of Warkentin et al. [61] and Botha et al. [66]. Our results partially contradict those of Mitchell et al. [64] and Liu et al. [65]. The desire to study face-to-face comes from young people’s need for interaction, connection, and feedback. In the current period, as the pandemic seems to continue, the measures for the protection of students, teachers, and the wider community must be correlated with young people’s need for real-life socialisation in order to avoid affecting their wellbeing, energy, and intellectual tone.

For the feeling of anxiety as a negative perception associated with personal development, our results are similar to those of Brown et al. [72] and Paechter and Maier [74], but contradict those of Burgess and Sieverstsen [73] and Hakawah [70]. The online teaching system had a negative effect on students’ perceptions of their personal development. They were worried that they would not be able to successfully complete their studies, and that they would not be able to take their exams in time. Contrary to our findings, Adesina and Orija [105] found that students perceived five major benefits of online learning: career advancement, scheduling flexibility, self-paced learning, a broader global perspective, and skill development.

For the decrease in students’ wellbeing as a result of their negative perceptions of the possibilities for personal development, our conclusions are similar to those reached by the UNDP [53], ECLAC–UNESCO [77], Sundarasen et al. [78], and Aristovnik et al. [82]. The way in which students perceive personal development in the context of the COVID-19 pandemic leaves deep marks on the development of their academic careers and their wellbeing. The uncertainty and anxiety about personal and professional futures lead to stressful situations for students and graduates, as also shown by Capone et al. [106].

The hypothesis that correlates students’ ease of adapting to online education with their perceptions of universities’ efficiency was validated, confirming the authors’ initial assumption that universities’ efficiency makes students perceive online education as easy, and with benefits for their professional and personal futures [93,94,95]. Flexibility, profitability, research perspectives in the online environment; internet network accessibility; and the interfaces of the platforms used were perceived as positive. Teachers’ delayed feedback; the lack of immediate technical support; the lack of self-regulation and motivation; monotonous, boring teaching methods; the strong sense of isolation; and poorly designed and unattractive teaching content were perceived as negative.

It turned out to be true that students’ confidence in their universities’ efficiency nullifies the feelings of anxiety and isolation. These results are in consonance with those of Basilaia and Kvavadze [83], UNESCO [95], and Mishra [88], but partially contradict those of Kibritchi et al. [91]. The positive perception of universities’ efficiency by both students and society has a significant positive effect on students’ wellbeing. No studies deny the importance of a positive perception. Moreover, the management and development strategies of any university pay special attention to the institution’s image, which subsequently translates into better visibility and higher ranking.

Both the validated (H3, H4, H5, H6, and H7) and non-validated (H1 and H2) hypotheses show that students experienced various situations and feelings that affected their wellbeing and, in many cases, their academic results as well.

Enriching the findings of this study through future research would be a particularly useful approach in order to find solutions for the academic community and the educational system.

## 6. Conclusions, Limitations and Future Research Directions

This article discusses a current and very important topic, focusing on the analysis of students’ wellbeing in the context of the isolation caused by the COVID-19 pandemic. The effects of online education during the COVID-19 pandemic on students’ wellbeing were analysed. The authors’ general perspective has been confirmed, and our findings corroborate those of Bali and Liu [33] and Platt, Raile, and Yu [34], showing that students have clear perceptions of online education, and consider it necessary for their career success. The relationship between online education as an unforeseen phenomenon and the feelings of isolation and anxiety requires increased attention from the university management.

Research shows that people with higher resilience levels are less anxious [35]. The threat of the pandemic has generated insecurity, fear, stress, vulnerability, and concern for the future, with negative impacts on wellbeing [35]. In the context of the pandemic, many people suffer from stress or anxiety, with fear of contagion, fear of the socioeconomic consequences of the pandemic, fear of foreigners because they may carry the disease (disease-related xenophobia), traumatic stress symptoms (nightmares, intrusive thoughts), etc. [39].

Women, young people, people living alone, people with health issues, people concerned with family members’ health, and those who feel discriminated against suffer from greater psychological impacts, leading to increased stress, anxiety, and depression. Conversely, perceptions of good health and trust in doctors have been significantly associated with a low psychological impact of the pandemic, and with reduced stress, anxiety, and depression [40].

People with stable incomes, people living with their families, and people who received social support suffered less from anxiety. In contrast, young people who had known other persons infected with COVID-19 were more anxious [41]. All of this confirms the connection between the uncertainty caused by the pandemic, on the one hand, and academic performance and professional development, on the other.

One positive aspect of social isolation is that it reduces the fear of contagion. Fear of contagion decreases as a result of isolation, but leaves room for anxiety, which worsens as the period of isolation increases. Individuals living with their families had a greater sense of security and protection, and a lower level of fear and vulnerability, with social support being very important in maintaining their emotional balance [43].

Students are acutely aware of the effects of the pandemic on the economy, their families, and society. The COVID-19 pandemic has caused an economic crisis, with severe effects on people’s wellbeing. Social distancing, isolation, and travel restrictions have reduced the labour force in all fields, and have caused widespread losses of jobs, resulting in a high level of fear regarding the imminence of a significant economic crisis [51]. Furthermore, schools were closed and the demand for supplies decreased—except for the food industry, with an increase in demand as a result of the panic that urged people to stock up on food.

The most powerful negative economic effects will be seen in production, agriculture, tourism, commerce, and industry, and the greatest economic issue in Europe will be unemployment [52]. The uncertain economic situation is a factor of stress and anxiety not only for employees but also for students, according to the relevant literature. The education sector is affected on all levels, from pre-school to tertiary education. States around the world have introduced various measures, ranging from permissive ones—where pupils are allowed to go to school, and certain professional groups are allowed to work from home—to radical measures, meaning the full closure of schools, extension of working from home, and even the prohibition of some activities.

Educational institutions have adapted to a brand new situation, and have applied new methods—some of them innovative—to implement online teaching with as few shortcomings as possible. Because of the pandemic, online education communities have developed, reflecting both an increase in and diversification of the student population, as well as the need for teachers to update their teaching skills, practices, and strategies to replace traditional classrooms with virtual ones.

Depending on the abilities of each student, their strength and desire for learning and training, and the way in which they invested their time in personal development before the pandemic, some students have been able to cope more easily with the changes. Personal development is essential in building a career. Both universities and students should be able to adapt and be open to the current changes. Practice has shown that a large part of the initial reluctance manifested by students is now gone, and the usual counselling and personal development meetings have been replaced by webinars, interactive workshops, and online conferences or counselling sessions, the results of which are as good as before the pandemic. In addition to what is already mentioned above, face-to-face learning and counselling—in which a shared understanding has to be derived, or in which interpersonal relations must be established—are preferred by students for communication purposes [74].

For a young person, the main support after the family is the educational environment. Schools are fundamental areas for emotional support, risk monitoring, and educational continuity; thus, maintaining emotional, psychological, and social wellbeing, as well as personal development, is a challenge for all of the members of the educational community: students, families, and teachers. Isolation and online education have put an end to the personal, face-to-face relationships between students and teachers, and within student groups. Our findings on students’ perceptions of the deterioration of social relationships can complete the database for educational policy.

Educational policy must aim not only at academic performance, but also at cultivating the elements that the graduate will need throughout their life for the most efficient management of crisis situations, such as the current pandemic. Individual wellbeing is one of the most important goals pursued by individuals, and derives from the basis of individual personal development [81]. Positive variations in wellbeing are materialised both in the easier integration of young graduates to the labour market and, directly, in the proper management of feelings and reactions under stress, including the pandemic, panic, feelings of isolation, and other negative feelings.

Parents, teachers, and educational institutions have attempted not only to facilitate learning, but also to provide social and moral support, and ensure interaction during school closures. Moreover, they have provided support and assistance in using e-learning, online, or phone counselling platforms [22,86,87].

Parents, school, and society shape young people’s perceptions of the world. High aspirations, care and positive attitude towards learning, motivation, and encouragement of group members (teacher–student or student–student) can all contribute to obtaining good academic results. Students who do not receive significant support from their families compensate for with the moral and social support provided by teachers and their peers, and can still attain good school results and be successful in general [88].

Traditional learning takes place face-to-face, where teachers and pupils or students are physically present. Not all pupils or students learn and adapt in the same way; almost two decades ago, it was found that the traditional approach was not suitable for all learners [88,89,90].

Online education is changing all elements of teaching and learning in higher education. Content transfer from traditional teaching to online teaching has generated problems and differences in perception [91]. Some students consider online courses ineffective because of the limited student interaction with the academic staff and colleagues, and the increased interaction with technology. Higher education institutions need to improve their teaching and learning strategies by analysing their students’ feedback [92]. The lack of control over students and the inability to solve this shortcoming has been widely felt; teachers have had to deal with how to deliver online content, and the decline in students’ interest in learning [91]. Regarding the efficiency of online learning, there are both positive and negative opinions—especially from teachers—as presented in the literature review [93,94]. Students’ opinions are very important, and they should be the subject of future research to accurately identify the needs of students in the context of the sudden transition to online learning.

In addition to the quality of information they receive in class, students also have other needs and expectations of their universities. Equally important are their perceptions regarding all of the measures taken by universities, their efficiency, their constant concern for the students’ wellbeing and the results they obtain, their constant requests for students’ feedback, the integration of their opinions and recommendations in university policies, adaptation to the current pandemic context, and their provision material (scholarships), moral (counselling), and technical support, making quality material and information resources available to students. All of the above strengthen students’ trust in the university and improve their wellbeing.

The access to specialised documentary materials, and to specific counselling and guidance activities suitable for each academic discipline, together with tutoring classes, are the major challenges to be overcome by universities with respect to online education. A new reality has emerged, where digital skills have achieved outstanding results among students and teachers; society has adapted to the new requirements, and what seemed almost inconceivable two years ago has become a norm, judging by how smoothly the online teaching activities are being carried out.

Depending on the measures applied to prevent and combat the pandemic so far, the pandemic has manifested itself differently from one country to another. However, for the educational environment, it is extremely important to have directions aimed at protecting young students and their health, such as providing equal and non-discriminatory educational opportunities by training all students equally, protecting students with vulnerabilities, and taking advantage of the opportunities to reorganise the training–educational process. One of the priorities for reorganising the education system is to build teams that include students, teachers, and people from outside the education system, in order to identify answers to current requirements [107,108].

Because it has been implemented under the pressure of circumstances, online or partially online education may seem too difficult; however, it can present an opportunity as long as we all manage to adapt to the new context. Online teaching has generated new experiences, for which the transfer of activity planning from an onsite to an online format is necessary. This change is considered to be a temporary one; the current goal is to allow the temporary online access to the content of the mandatory university syllabuses, not the creation of a new education system [30].

The degree of students’ satisfaction with online education cannot be researched in a unitary way for all countries. In Romania, online education is less developed compared to other states. As a result, the satisfaction of Romanian students is high, because of the novelty and flexibility specific to online learning. However, to the same extent, there is a certain degree of dissatisfaction due to the lack of direct interaction, and this is something rather new to Romanian students, as they now bear the direct responsibility for accessing the courses. As the academic staff adapt their learning methods according to the students’ cognitive and emotional needs, their wellbeing may improve. The present situation means that online learning is not an option, but a necessity in most cases. Logistical weaknesses, insufficient pedagogical adaptation, and lack of direct interaction with colleagues and teachers reduce students’ wellbeing and increase the desire for onsite education. In order to increase wellbeing in the context of online education, the focus of universities will have to be on how to enable social–emotional learning in virtual classrooms. Thus, educational institutions should adapt rapidly and identify solutions so that their students will be able to achieve wellbeing, even if they are not able to refer to a specific model [67].

For the time being, there are no relevant studies in Romania regarding students’ perceptions of the pandemic and its influence on their wellbeing.

The empirical results obtained so far show that Romanian students tend to have a positive perception and a good mental state under the conditions of the traditional education system. An economic crisis does not seem to diminish the wellbeing of Romanian students, even if the future may seem insecure. Fear of contagion is low, and does not cause concern to young people; thus, the effect on their wellbeing is neutral. Romanian students seem to have a positive perception of the effects of online education on their personal development, such that their wellbeing is not affected. The capability of higher education institutions to purchase and efficiently use the infrastructure necessary to the online system, as well as to adapt to this system, is perceived as a form of institutional efficiency, which allows the correct conveying and receiving of information. This increases students’ trust and, thus, their wellbeing.

This study has several limitations. The respondents came from Romanian universities, and most of them (80.45%) were female students. However, the sample is representative, as the majority of students in the universities and specialisations under study are women. A wider institutional range would provide a more complex picture of the perceptions of young people studying in the Romanian university system with regard to online teaching and learning. This study was limited to Romanian students, and did not analyse, by comparison, the perceptions of young people studying in other countries. An analysis of the teachers’ perceptions of the same questions in the research hypotheses would reflect the views of groups with different roles, but with common goals within the same process. Another limitation derives from the received answers, in the sense that they do not provide any certainty about the students’ perceptions regarding certain aspects. For instance, the low fear of contagion may be the result of insufficient information or immaturity, which can also influence the perception of personal development. Another limitation is the evolution of the pandemic, as the rate of decline in the number of infections is unpredictable. Education is one of the most immediately affected sectors judging by the current shortcomings in online learning, as well as the long-term losses in terms of students’ knowledge and skills, according to the requirements of the labour market. Research is necessary to identify predictability models to prepare the future reactions in online learning, to maximise the advantages of online education, and to properly manage the inherent shortcomings from the personal and emotional points of view.

Surprisingly, the COVID-19 pandemic’s effects on education have been both positive and negative. Educational institutions have adapted to the current situation, and have identified ways to continue their activities under conditions that are very similar to those before the pandemic. To a great extent, the past two years have brought positive changes in the paradigm of the teaching and assessment process. The negative aspect is related to students’ lack of interaction, with possible negative emotional consequences. The relevant entities will have to take a closer look at supporting the education sector by counselling young people in the field of personal and emotional development.

Our results show that Romanian students trust their ability to develop within the online system, but their optimism can be the result of an erroneous self-assessment. This perception could have been influenced by the inconsistent communication strategy practiced by the line ministries (the Ministry of National Education and the Ministry of Public Health) during the pandemic [109].

The timing of the questionnaire may be another limitation of the research, as young people were still adjusting to the transition from the traditional to the online system, without having fully adapted to the new teaching/learning methods. In addition, the results were not clear, i.e., the grades obtained after the online exams represented their performance over a short time. Resuming the questionnaire after a certain time and using the same sample of respondents could lead to different results. Such limitations open new directions of research, with questionnaires adapted to new realities and applied to larger samples, i.e., the respondents could include students from other universities or other countries, or even the inclusion of other impact groups, such as teachers or parents.

This study complements the literature while offering, in a relative manner, the perceptions of Romanian students on the implementation of online education during the pandemic. The results are a starting point in designing a future strategy in education at the national and institutional levels, as there could be a hybrid system in the future, and the return to the traditional education system may not happen.

## Figures and Tables

**Figure 1 ijerph-18-09665-f001:**
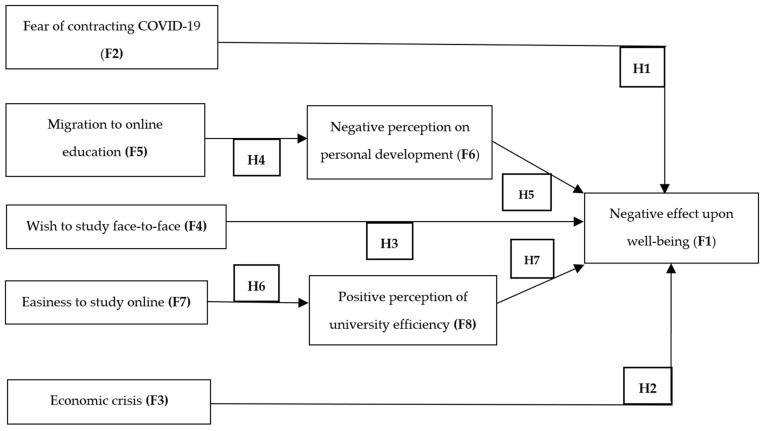
The structural model (F: factor).

**Figure 2 ijerph-18-09665-f002:**
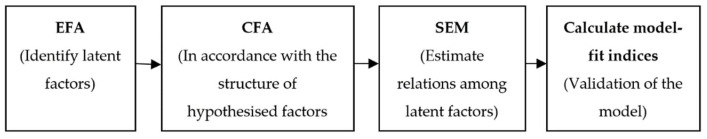
SEM modelling design. Note: EFA: Exploratory Factor Analysis; CFA: Confirmatory Factor Analysis; SEM: Structural Equation Modelling.

**Figure 3 ijerph-18-09665-f003:**
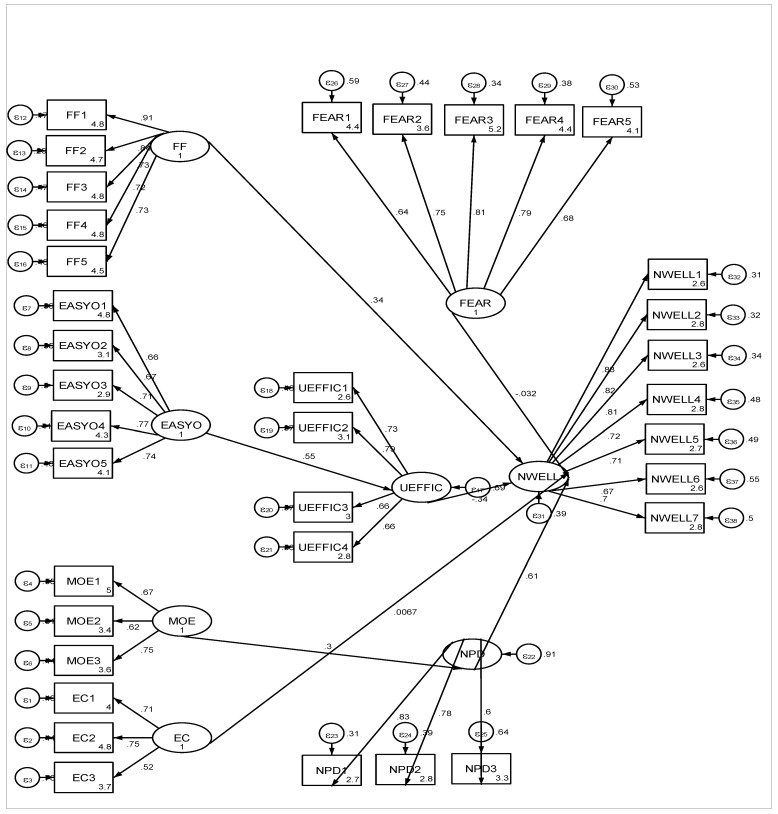
Results of the structural model.

**Table 1 ijerph-18-09665-t001:** Descriptive statistics.

Descriptive Statistics			Number	Percentage (%)
Gender	Male		130	19.55
	Female		535	80.45
Education level	Undergraduate students	1st year	71	10.67
		2nd year	269	40.45
		3rd year	168	25.26
	Postgraduate students (master’s studies)	1st year	65	9.77
		2nd year	92	13.84

Source: authors’ calculations based on Stata statistical analysis software.

**Table 2 ijerph-18-09665-t002:** The values of the Cronbach’s alpha index for the analysed factors and the variable validity testing.

Latent Variable (Factors)	Questions/Observed Variable	Cronbach’s Alpha	Loading (std)	Construct Reliability (CR)
Negative effects on students’ wellbeing (NWELL) F1	I feel like I can’t deal with online learning.	0.9108	0.832	0.903
	2. Learning at home demotivates me.		0.823	
	3. I can’t concentrate during online courses.		0.811	
	4. I feel more stressed since the courses take place online.		0.722	
	5. Since I’ve been in isolation, I have been much more inefficient in the learning process.		0.713	
	6. Since I’ve been in isolation, I have not been able to organize my learning time effectively.		0.672	
	7. This period will have a negative effect on my school performance in the future.		0.705	
Fear of contracting the virus (FEAR) F2	1. How worried are you that you or a family member will be exposed to SARS-CoV-2?	0.8513	0.748	0.684
	2. I am worried that I may contract SARS-CoV-2.		0.810	
	3. I am worried that someone in my family may contract SARS-CoV-2.		0.788	
	4. I am worried that I could spread SARS-CoV-2.		0.716	
	5. I am worried that a cure for the virus will not be found very soon.		0.684	
Economic crisis (EC) F3	1. I think the unemployment rate will increase next year.	0.6887	0.713	0.702
	2. I think there will be an economic crisis next year.		0.745	
	3. I try to save as much money as possible for the next period.		0.523	
Face-to-face education (FF) F4	1. I want to go to classes in a real classroom.	0.8995	0.910	0.891
	2. I want to see my teachers again.		0.840	
	3. I want to see my classmates again.		0.726	
	4. I feel that being present in class is mandatory for an efficient learning.		0.718	
	5. I want to go back to college as soon as possible.		0.734	
Migration to online education (MOE) F5	1. The COVID-19 pandemic urged the natural migration of education to online.	0.7209	0.668	0.722
	2. The COVID-19 pandemic will influence the education system by determining change through surprising innovations.		0.624	
	3. Universities will adapt as a result of the crisis determined by SARS-CoV-2.		0.750	
Negative perception of personal development (NPD) F6	1. This period will have a negative effect on my career.	0.7747	0.830	0.830
	2. This period will have a negative effect on my personal development.		0.781	
	3. There will be less students willing to sign up for an admission exam at one of the universities abroad.		0.597	
Ease of studying online (EASYO) F7	1. I am able to easily use the Internet according to my education interests.	0.8311	0.660	0.835
	2. I am used to online communication.		0.671	
	3. I could easily adapt to online courses without the direct assistance of my teachers.		0.710	
	4. I am willing to communicate actively online with my classmates and teachers.		0.769	
	5. I am capable of self-discipline and find time for study at home.		0.735	
Positive perception of university efficiency (UEFFIC) F8	1. The university was prepared to manage online teaching.	0.8051	0.730	0.805
	2. Teachers manage to effectively convey the main ideas and knowledge to students through online programs.		0.791	
	3. The university will be able to guarantee my safety in the future.		0.660	
	4. Online communication with teachers was smooth.		0.664	

Source: authors’ elaboration based on Stata statistical analysis software. Note: NWELL: Negative effects on students’ wellbeing; FEAR: Fear of contracting the virus; EC: Economic crisis; FF: Face-to-face education; MOE: Migration to online education; NPD: Negative perception of personal development; EASYO; Ease of studying online; UEFFIC: Positive perception of university efficiency.

**Table 3 ijerph-18-09665-t003:** Acceptable fit indices for the CFA and the measurement model.

RMSEA	CFI	TLI	SRMR
<0.100	>0.900	>0.900	<0.09

Note: RMSEA: Root Mean Square Error of Approximation; CFI: Comparative Fit Index; TLI: Tucker-Lewis Index; SRMR: Standardized Root Mean Squared Residual. Source: Dragan and Topolšek [99].

**Table 4 ijerph-18-09665-t004:** The results of the estimations.

Hypotheses		Parametre Estimations	
		Coefficient	Is the Hypothesis Supported?
H1	FEAR→NWELL	−0.031 (0.358)	NO
H2	EC→NWELL	0.006 (0.855)	NO
H3	FF→NWELL	0.337 *** (0)	YES
H4	MOE→NPD	0.301 *** (0)	YES
H5	NPD→NWELL	0.611 *** (0)	YES
H6	EASYO→UEFFIC	0.554 *** (0)	YES
H7	UEFFIC→NWELL	−0.340 *** (0)	YES

Note: *** represent *p* < 0.01. Source: authors’ calculations using Stata statistical analysis software version 13. NWELL: Negative effects on students’ wellbeing; FEAR: Fear of contracting the virus; EC: Economic crisis; FF: Face-to-face education; MOE: Migration to online education; NPD: Negative perception of personal development; EASYO; Ease of studying online; UEFFIC: Positive perception of university efficiency.

## Data Availability

Not applicable.

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
