# Peer review of "Students’ Wellbeing during Transition from Onsite to Online Education: Are There Risks Arising from Social Isolation?"

_ijerph, 2021, doi:10.3390/ijerph18189665_

Round 1
Reviewer 1 Report
Thank you for the opportunity to evaluate the article "Students Well-Being as a Result of Education through On-site to On-line. Are there Risks Arising from Social Isolation?" First, I would like to congratulate the authors for having addressed a current issue that requires immediate research to support some type of action that allows improving of adolescents´ well-being. Therefore, I appreciate the effort made by the authors to cover a relevant topic that requires further scientific research. This is an article with great potential, but one that, in my humble opinion, requires greater precision in its writing. In consequence, the recommendations made are intended to help authors improve the article, due to its potential and topic of interest:
INTRODUCTION
- The authors have made a great effort to support the hypotheses. However, the writing of the theoretical framework is too extensive. In this sense, I do not think it is a good idea to describe the jobs one by one. In fact, I think it would be more appropriate to support the reality that they intend to convey through prior research. In this way, the authors would avoid the many redundancies that occur during the development of this section.
- In fact, many of the contributions made in the theoretical framework could be used to develop more consistent conclusions in their respective section, which would notably enrich the conclusions section.
- Also along the theoretical framework, cross-sectional statements are made, such as the influence of “moral support” (line 382) or “resilience” (line 113) (among others) that should be avoided. These explanations will probably make more sense in the conclusions, in case the authors intend to justify some result or dynamics of the analyzed relationships.
- In my opinion, it is not clear which are the variables analyzed in the work. The authors do not provide definitions of what they call "adolescent well-being", "personal development", etc.
In short, the theoretical framework is somewhat confusing and the objectives of the study are not explicitly mentioned either. The variables should be defined with greater precision and much of the information given could be incorporated in the conclusions, which would enrich this last section.
METHOD
- The authors describe aspects concerning the method in the results section. Statistical procedures and cut-off points could be included in a new section called “statistical analysis”. Likewise, the authors do not specify which measurement instrument was used. Therefore, the authors could create a new section called “instruments” and justify the validity and reliability in this section.
- In addition, the authors should avoid information that, in my opinion, is not necessary: the formula for Alpha, CR ...
- The authors should report information on the normality of the sample to justify the selected analysis.
RESULTS
- The authors mistakenly use the term item in the tables
- Authors should specify whether the coefficients are standardized or non-standardized.
- The authors do not include the indices obtained in the measurement model, a prerequisite for checking the structural model. They also do not include a correlational analysis. Therefore, the authors should include all this information.
- In the description of the results referring to each hypothesis, the authors include statements that are specific to the conclusions section (for example: Our conclusions are in consonance with those of Basilaia and Kvavadze [81], UNESCO [100] and Mishra [86], but they contradict to a certain extent those of Kibritchi et al. [89]). Therefore, I recommend that the authors only include information referring to the results.
- The content of tables 2 and 4 could be expressed in a single one.
CONCLUSIONS
In my opinion, this section could be substantially enriched by justifying the results obtained in the study, for which the authors can use the theoretical framework.
OTHERS
Please revise: “herefore” (line 549); coefficients “p”, “β”… should be expressed in cursive.
Author Response
Ref: IJERPH - art.no. 1279295
Point-by-Point Response to Reviewers
Regarding Reviewer Report of: Students Well-Being as a Result of Education through On-site to On-line. Are there Risks Arising from Social Isolation?
Notes: We reproduce the original comments in italics for reference and respond beneath each point.
Response to Reviewer 1
Comment 1: Thank you for the opportunity to evaluate the article "Students Well-Being as a Result of Education through On-site to On-line. Are there Risks Arising from Social Isolation?" First, I would like to congratulate the authors for having addressed a current issue that requires immediate research to support some type of action that allows improving of adolescents´ well-being. Therefore, I appreciate the effort made by the authors to cover a relevant topic that requires further scientific research. This is an article with great potential, but one that, in my humble opinion, requires greater precision in its writing. In consequence, the recommendations made are intended to help authors improve the article, due to its potential and topic of interest:
Response to comment 1: The authors would like to thank you for your feedback!
Comment 2: The authors have made a great effort to support the hypotheses. However, the writing of the theoretical framework is too extensive. In this sense, I do not think it is a good idea to describe the jobs one by one. In fact, I think it would be more appropriate to support the reality that they intend to convey through prior research. In this way, the authors would avoid the many redundancies that occur during the development of this section.
Response to comment 2: Thank you for your remarks. The theoretical part was indeed extensive. Consequently, we gave up some explanations and included them in the conclusions. We maintained the structure of the paper. The hypotheses were grouped together and presented immediately after the conceptual model of the study.
Comment 3: In fact, many of the contributions made in the theoretical framework could be used to develop more consistent conclusions in their respective section, which would notably enrich the conclusions section.
Response to comment 3: Thank you for your remarks. The text that was deleted from the theoretical part was included in the conclusions.
Comment 4: Also along the theoretical framework, cross-sectional statements are made, such as the influence of “moral support” (line 382) or “resilience” (line 113) (among others) that should be avoided. These explanations will probably make more sense in the conclusions, in case the authors intend to justify some result or dynamics of the analyzed relationships.
Response to comment 4: Thank you for your remarks. We have included the statements regarding "moral support" and "resilience" in the conclusions.
Comment 5: In my opinion, it is not clear which are the variables analyzed in the work. The authors do not provide definitions of what they call "adolescent well-being", "personal development", etc.
Response to comment 5: Thank you for your remarks. We avoided using the concept of "adolescent well-being" because, according to Ross (2020), it only refers to people under 18 years of age. But we defined, in the Introduction, the concept of well-being and adapted it to our target group, the students. The source of definition is Mansfield Louise, Daykin Norma, Kay Tess (2020). We also defined the concept of "personal development" in part 2.4 (Online education and students’ personal development), based on Lejeune et al. (2018).
Ross, David et al. (2020). Adolescent Well-Being: A Definition and Conceptual Framework. Journal of Adolescent Health, 67 (2020) 472e476
Mansfield Louise, Daykin Norma, Kay Tess (2020). Leisure and Wellbeing. Leisure Studies, 39(1): 1-10. https:doi.org/10.1080/02614367.2020.1713195.
Lejeune Christophe, Beausaert Simon, Raemdonck Isabel (2018): The impact on employees’ job performance of exercising self-directed learning within personal development plan practice, The International Journal of Human Resource Management, DOI: 10.1080/09585192.2018.1510848
Comment 6: In short, the theoretical framework is somewhat confusing and the objectives of the study are not explicitly mentioned either. The variables should be defined with greater precision and much of the information given could be incorporated in the conclusions, which would enrich this last section.
Response to comment 6: Thank you for your remarks. We specified, in the Introduction, the main objective of the research and we developed the conclusions including information from the theoretical part.
Comment 7: The authors describe aspects concerning the method in the results section. Statistical procedures and cut-off points could be included in a new section called “statistical analysis”. Likewise, the authors do not specify which measurement instrument was used. Therefore, the authors could create a new section called “instruments” and justify the validity and reliability in this section.
Response to comment 7: Thank you for your remarks. We added a new section called Statistical analysis which presents the analyses used in the article. We employed the SEM (Structural Equation Modelling) technique, which, according to Dragan and Topolsek, is split into 4 work phases:
- a) Exploratory factor analysis;
- b) Confirmatory factor analysis (measurement model);
- c) The estimation of the relations among the latent factors (structural model);
- d) Validation of the model.
Dragan, D.; Topolšek, D. Introduction to Structural Equation Modeling: Review, Methodology and Practical Applications. In The International Conference on Logistics & Sustainable Transport, Celje, Slovenia, 19–21 June 2014
Comment 8: In addition, the authors should avoid information that, in my opinion, is not necessary: the formula for Alpha, CR ...
Response to comment 8: Thank you for your remarks. We gave up some information that the reviewers did not consider important: the formula for Cronbach Alpha, CR, etc.
Comment 9: The authors should report information on the normality of the sample to justify the selected analysis.
Response to comment 9: Thank you for your remarks. We used ordinal 5-point Likert scale data. So, by definition our data are not normally distributed. But we use statistical tests which do not make assumptions about the distribution of the data, i.e., nonparametric or distribution-free tests. We believe that the SEM technique doesn’t assume normality. Also, central limit theorem states that when sample size has 100 or more observations, violation of the normality is not a major issue.
Comment 10: The authors mistakenly use the term item in the tables
Response to comment 10: Thank you for your remarks. We gave up the word "item" and replaced it with "factor".
Comment 11: Authors should specify whether the coefficients are standardized or non-standardized.
Response to comment 11: Thank you for your remarks. We added that the coefficients are standardized.
Comment 12: The authors do not include the indices obtained in the measurement model, a prerequisite for checking the structural model. They also do not include a correlational analysis. Therefore, the authors should include all this information.
Response to comment 12: Thank you for your remarks. We included the indices obtained in the measurement model, such as RMSEA, CFI, TLI and SRMR.
Comment 13: In the description of the results referring to each hypothesis, the authors include statements that are specific to the conclusions section (for example: Our conclusions are in consonance with those of Basilaia and Kvavadze [81], UNESCO [100] and Mishra [86], but they contradict to a certain extent those of Kibritchi et al. [89]). Therefore, I recommend that the authors only include information referring to the results.
Response to comment 13: Thank you for your remarks. We tried to eliminate the phrases that refer to conclusions and comparisons to other authors from our results section.
Comment 14: The content of tables 2 and 4 could be expressed in a single one.
Response to comment 14: Thank you for your remarks. In the revised manuscript, we merged Tables 2 and 4. A single table resulted (Table 2), and Table 4 was deleted.
Comment 15: In my opinion, this section could be substantially enriched by justifying the results obtained in the study, for which the authors can use the theoretical framework.
Response to comment 15: Thank you for your remarks. In the revised manuscript, we used fragments from the theoretical part to develop the conclusions.
Comment 16: Please revise: “herefore” (line 549); coefficients “p”, “β”… should be expressed in cursive.
Response to comment 16: Thank you for your remarks. "Herefore" is a typographical error for "therefore" and coefficients “p”, “β”… have been expressed in cursive. We’ve made the changes.
We mention that we slightly changed the title of the paper, this being necessary.
Thank you again for the suggestions and the time allocated to reading our paper.
The authors

Reviewer 2 Report
I wish to congratulate the authors for the work done. The paper is very interesting and very well written.
But I have few remarks that must be answered/resolved:
1) I didn't find in the text what survey was used? Who constructed this survey and was there any test-retest reliability. This is crucial to any survey-based investigation.
2) Why was there such disproportion of sexes (females 80%)?
3) Separate results and discussion sections. Now, it is kind of confusing to the non-expert and ordinary readers.
4) Part "Literature review and research hypotheses" could be shorter, but this is just a recommendation.
5) Please clearly state what statistical methods/analyses were used in the new section named Statistical analysis. Work phases are just part of a procedures
Author Response
Ref: IJERPH - art.no. 1279295
Point-by-Point Response to Reviewers
Regarding Reviewer Report of: Students Well-Being as a Result of Education through On-site to On-line. Are there Risks Arising from Social Isolation?
Notes: We reproduce the original comments in italics for reference and respond beneath each point.
Response to Reviewer 2
Comment 1: I wish to congratulate the authors for the work done. The paper is very interesting and very well written.
Response to comment 1: The authors would like to thank you for your feedback!
But I have few remarks that must be answered/resolved:
Comment 2: I didn't find in the text what survey was used? Who constructed this survey and was there any test-retest reliability. This is crucial to any survey-based investigation.
Response to comment 2: Thank you very much for these recommendations. We used a survey constructed by other authors and already published. We din not displayed our survey because it is very long, with many sections. Reliability of the survey was confirmed by our exploratory factor analysis.
Comment 3: Why was there such disproportion of sexes (females 80%)?
Response to comment 3: Thank you very much for these recommendations. This study has several limitations. The respondents, most of them (80.45%) are female students. However, it is indeed a representative sample, as this female majority is real in the universities and specializations we selected for the research.
Comment 4: Separate results and discussion sections. Now, it is kind of confusing to the non-expert and ordinary readers.
Response to comment 4: Thank you for your valuable suggestion. We simplified the sections that now only include conclusions, limitations and future research directions. Discussions are now sepparate.
Comment 5: Part "Literature review and research hypotheses" could be shorter, but this is just a recommendation.
Response to comment 5: Indeed, "Literature review and research hypotheses" was extensive. Consequently, we gave up some explanations and included them in the conclusions.
Comment 6: Please clearly state what statistical methods/analyses were used in the new section named Statistical analysis. Work phases are just part of a procedures
Response to comment 6: Thank you for your remarks. In the revised manuscript, we added a new section called Statistical analysis which presents the analyses used in the article. We employed the SEM (Structural Equation Modelling) technique, which, according to Dragan and Topolsek, is split into 4 work phases.
- a) Exploratory factor analysis;
- b) Confirmatory factor analysis (measurement model);
- c) The estimation of the relations among the latent factors (structural model);
- d) Validation of the model.
Dragan, D.; Topolšek, D. Introduction to Structural Equation Modeling: Review, Methodology and Practical Applications. In The International Conference on Logistics & Sustainable Transport, Celje, Slovenia, 19–21 June 2014
Thank you again for the suggestions and the time allocated to reading our paper.
The authors

Round 2
Reviewer 1 Report
First of all, I would like to congratulate the authors for the effort made in reviewing their article, since they have included important statistical analyzes and have also clarified the various sections of the present work. However, in my humble opinion the authors could take into considerations the next suggestions:
I would avoid the next paragraph form the introduction: “This paper is organised in the following sections: introduction, analysis of specific literature, and development of research assumptions, research methodology, results, discussions and conclusions” (section 1).
I think that it would be more appropriate to move the paragraph “This study focused on the following issues: students’ well-being; their willingness to study in the traditional way; how they relate to online education; contamination fear; the role of universities in creating a safe environment; students’ perception of the impact on their career and personal journey; the evolution of the education system in the future; economic implications of the COVID-19 pandemic.” to the last one behind “This study is a comparative analysis of the effect on the well-being of students from universities in Romania in the context of the online education imposed by the COVID-19 pandemic. The perception of students and pupils on the effectiveness of online courses has been studied intensely by Bentley [32], Bali and Liu [33], Platt, Raile and Yu [34]. However, they all used a hybrid scenario, with both onsite and online courses. The novelty of this research is that it analyse the impact of the pandemic on the well-being of the students studying for bachelor’s and master’s degree, where only online courses were available.”
Maybe the authors could create a 2.7 section named “the present study” in which they may explain in short the relevancy and need of their study. Also, should include the objective of the present work (to prove a theoretical model about….) and present, as they have done, the figure and their respective hypotheses.
I would delete from discussion the statement: “As mentioned in the beginning of the article” (line 1).
In the discussion section the authors enumerate each hypothesis. It would be convenient to include ":" in the previous paragraph or at some point any other element of cohesion.
It is quite difficult to follow the common thread of section (6. Conclusions, limitations and future research directions). It would be convenient for the authors to include cohesion elements, since it gives the impression that the authors list single paragraphs without any relationship. Indeed, it is an interesting and complex study, which analyzes the relationship between several variables through sem. Therefore, and precisely for this reason, it is very important that the wording is clear and follows a clear common thread. An aspect that in this last section can be improved.
Author Response
Ref: IJERPH - art.no. 1279295
Point-by-Point Response to Reviewers
Regarding Reviewer Report of: Students’ Well-Being as a Result of the Transition from Onsite to Online Education. Are there Risks Arising from Social Isolation?
Notes: We reproduce the original comments in italics for reference and respond beneath each point.
Response to Reviewer 1
Comment 1: First of all, I would like to congratulate the authors for the effort made in reviewing their article, since they have included important statistical analyzes and have also clarified the various sections of the present work.
Response to comment 1: The authors would like to thank you for your feedback!
However, in my humble opinion the authors could take into considerations the next suggestions:
Comment 2: I would avoid the next paragraph form the introduction: “This paper is organised in the following sections: introduction, analysis of specific literature, and development of research assumptions, research methodology, results, discussions and conclusions” (section 1).
Response to comment 2: Thank you for your suggestion. As recommended, we have removed the paragraph from the introduction.
Comment 3: I think that it would be more appropriate to move the paragraph “This study focused on the following issues: students’ well-being; their willingness to study in the traditional way; how they relate to online education; contamination fear; the role of universities in creating a safe environment; students’ perception of the impact on their career and personal journey; the evolution of the education system in the future; economic implications of the COVID-19 pandemic.” to the last one behind “This study is a comparative analysis of the effect on the well-being of students from universities in Romania in the context of the online education imposed by the COVID-19 pandemic. The perception of students and pupils on the effectiveness of online courses has been studied intensely by Bentley [32], Bali and Liu [33], Platt, Raile and Yu [34]. However, they all used a hybrid scenario, with both onsite and online courses. The novelty of this research is that it analyse the impact of the pandemic on the well-being of the students studying for bachelor’s and master’s degree, where only online courses were available.”
Response to comment 3: Thank you for your suggestion. The paragraph has been moved and now concludes, more appropriately, the introductory part of the new version of the article.
Comment 4: Maybe the authors could create a 2.7 section named “the present study” in which they may explain in short the relevancy and need of their study. Also, should include the objective of the present work (to prove a theoretical model about….) and present, as they have done, the figure and their respective hypotheses.
Response to comment 4: Thank you for your suggestion. A new section has been introduced, that is, 2.7. The present study, in which we briefly explained the importance and relevance of the paper, along with the model we used and the hypotheses. The new section indeed facilitates the presentation of the model in a clearer and more efficient way.
Comment 5: I would delete from discussion the statement: “As mentioned in the beginning of the article” (line 1).
Response to comment 5: Thank you for your suggestion. The statement has been deleted, as recommended.
Comment 6: In the discussion section the authors enumerate each hypothesis. It would be convenient to include ":" in the previous paragraph or at some point any other element of cohesion.
Response to comment 6: Thank you for your suggestion. We have given up the numerical listing of hypotheses and, instead, we have inserted specific cohesion sentences.
Comment 7: It is quite difficult to follow the common thread of section (6. Conclusions, limitations and future research directions). It would be convenient for the authors to include cohesion elements, since it gives the impression that the authors list single paragraphs without any relationship. Indeed, it is an interesting and complex study, which analyzes the relationship between several variables through sem. Therefore, and precisely for this reason, it is very important that the wording is clear and follows a clear common thread. An aspect that in this last section can be improved.
Response to comment 7: Thank you for your evaluation. We have now tried to provide a more visible logical thread in the final section of the paper, which more clearly orders the ideas that the authors wanted to convey.
Thank you again for the suggestions and the time dedicated to reading our paper.
The authors
